# Assessing Urban Flooding Extent of the Baunia Khal Watershed in Dhaka, Bangladesh

Zarin Subah [1], Sujit Kumar Bala [2] and Jae Hyeon Ryu [3,*]

1   College of Natural Resources, University of Idaho, Boise, ID 83702, USA
2   Institute of Water and Flood Management, Bangladesh University of Engineering and Technology, Dhaka 1000, Bangladesh
3   Soil and Water Systems, University of Idaho, Boise, ID 83702, USA
*   Correspondence: jryu@uidaho.edu

**Abstract:** Due to the gradual encroachment of natural drainage channels (also known as khals) and the poor management of drainage infrastructures, any rainfall of higher intensity for a few hours causes urban flooding in Dhaka, Bangladesh, hindering the daily lives of city dwellers, especially in slum areas. The Intensity–Duration–Frequency (IDF) curves with return intervals of 2, 5, 10, 25, 50, and 100 years were estimated for a 3 h duration of rainfall using the Gumbel statistical method to assess urban flooding extent around the Baganbari slum within the Baunia Khal watershed. The spatial land use changes of the Baunia Khal were also delineated through the analysis of the areal changes of the khal from 2004 to 2020. The area of the khal was found to be 7.00 km$^2$, 2.17 km$^2$ and 0.41 km$^2$ for the years 2004, 2010, and 2020, respectively. It appears that a gradual massive areal decrease of the khal (impervious land segment) was in progress during this period for high encroachments driven by various governmental agencies and other private entities' urban developmental projects. The runoff coefficient was computed as 0.76 based on the existing land use pattern, slope, and soil type of the study area. The average runoff generated from the catchment was estimated using the rational method and was 103.41 m$^3$/hr. The drainage length was calculated as 9.1 km using the Kirpich method, whereas the present drainage length was reported as only 6.95 km. The multiple rainfall intensities with return periods of 2, 5, 10, 25, 50, and 100 years were also applied to replicate a heightened extent of urban flooding in the Baunia Khal watershed. The study suggests that the depth, length and width of the Baunia Khal need to increase to hold the generated runoff to manage urban flooding around the Baunia Khal watershed.

**Keywords:** climate change; land use change; urban flood; khals

## 1. Introduction

The population of those living in urban areas is growing faster and has overtaken populations in rural areas, and it is predicted that urban populations will continue to rise globally [1]. As the majority of people move toward cities, the low-lying flood plains are converted into impervious areas for human habitation, leading to an improper infiltration process from raindrops to the soil column [2]. The impermeable soil accelerates large runoff events which inundate areas whenever significant rainfall occurs. The densely populated areas carrying high economic value often face massive impedance in urban development works due to flooding [3]. Among the disaster-prone countries, Bangladesh is ranked as the ninth-highest disaster-risk country in the world [4] and has a high population density. Almost 163 million people live in Bangladesh [5], and 19.6 million of those people live in its capital, Dhaka [6]. Dhaka, the mega city, struggles with flooding due to its topography and geolocation near the Ganges–Brahmaputra-Meghna system [7], a basin shared by four countries, Bhutan, Bangladesh, India, and Nepal. The embankments created after the 1998 flood helped to decrease riverine floods, but Dhaka faces frequent urban flooding during

the monsoon period. It depicts that urban flooding has a great impact on the city dwellers in Dhaka, especially the urban poor [8].

Urban flooding is a frequent phenomenon that happens when the drainage facilities of a city or town are not capable of draining water flows caused by natural or anthropogenic hydro events [9]. Effects of climate change and land use pattern changes have led to more frequent, longer, and higher floods which have deteriorated the situation of city dwellers [10,11]. Urban flooding brings about inevitable problems during the monsoon period (June–October) for the city dwellers in Dhaka [12]. Dhaka is the capital city of Bangladesh and is one of the most populous cities in the world. The city is intrinsically surrounded by a network of rivers for which the combined effect of upstream flow and monsoon rainfall leaves the city vulnerable to urban flooding. Even the low-frequency flood events that happened in Bangladesh in the past few decades have caused serious threats to the lives of people [13]. Riverine flooding has decreased over the past few decades, but high-intensity rainfall causes urban flooding during the monsoon period [14,15].

Climate change impacts urban flooding by increasing the number of occurrences and the intensity of rainfall [16,17]. The combined effect of climate change and urbanization is responsible for recent flooding [18,19]. The process that involves the warming of the environment makes the concentration of moisture in the atmosphere turn into intense rainfall events [20]. Consequently, the extreme heat over an urban heat island (UHI) with an enhanced aerosol load increases rainfall intensity, thereby leading to urban flooding [21]. Under the circumstances, flooding in Dhaka is likely taking place due to a small-intensity rainfall event where city dwellers often experience ankle-to-knee depth water on the roads [22].

Intensity–duration–frequency (IDF) curves are, therefore, necessary for designing water management and planning projects. Rainfall duration, frequency, and return periods are typically expressed using IDF curves on double logarithmic paper. Thus, Gumbel probability distribution [23] and parameter estimation methods are used to obtain IDF curves from statistical analysis of rainfall intensity. It provides a simple and practical procedure for predicting rainfall intensity and allows the estimation of the return period of an observed rainfall or vice versa [24]. It also helps to predict future extreme rainfall events in a region [25,26]. Design storms that are derived from IDF curves are frequently adopted for designing urban drainage systems, assessing flood vulnerabilities for a region, and evaluating hydraulic structure endurance [27].

The landform characteristic of Dhaka is the elevated Pleistocene terrace, which stands higher than the other neighboring low-lying marshland and floodplains [28]. In addition to this, the impermeable soil surface, solid waste dumping in the water bodies, and high-intensity rainfall increase urban flooding extent. A study showed that a 100-year return period flood could double in size, and a small flood event could be 10 times higher than it is if 30% of the roads in urban areas are paved [29]. Dhaka is predominantly characterized by limited land areas for physical development due to the surrounding wetlands, lowlands, and adjacent rivers [30]. The unauthorized construction and development on the banks of rivers, khals, and wetlands of the city increase urban flooding problems [31]. Illegal acquisition or encroachment to construct roads, houses, box culverts, etc. brings drastic changes by reducing the carrying capacity of the khals [32,33].

The selected study area, Baunia Khal, is vigorously encroached on due to urbanization and dwellers' practice of disposing of waste in slums [34]. A pair-wise ranking was carried out in Baganbari slum (situated alongside Baunia Khal) on the existing water-related problems associated with water supply, water price, urban flooding, water quality, and sanitation. Urban flooding came out as the main problem of the slum (see Table A1) for the slum dwellers. Therefore, the heightened extent of urban flooding also exacerbates gender suffering by creating immense stress on their livelihoods [35–37]. This study is conducted to generate the peak runoff events for different return periods of rainfall intensity in the Baunia Khal watershed and suggests the spatial changes that need to be taken to avoid the urban flooding hazard by meeting two objectives: (1) to analyze the spatial changes of the

Baunia Khal and generate peak runoff for extreme rainfall events, and (2) to investigate the extent of urban flooding in the Baunia Khal.

## 2. Study Area and Approach

The Baunia Khal watershed is located between latitudes 23°47′54.70″ N and 23°50′42.73″ N, longitudes 90°23′13.62″ and 90°21′47.67″ E, as shown in Figure 1. According to the "Storm Water Master Plan for Dhaka", the length of the Baunia Khal was about 7.63 km in 2016 and the flow direction of this khal was from east to west [38]. Many settlement and urban activities took place along the wetlands of the Baunia Khal through encroaching activities. The Baganbari slum is formed in the wetlands of the Baunia Khal in its Mirpur-14 section [39]. The area of the slum is 1.01 ha. The houses of the slum dwellers are made of poor-quality materials, such as bamboo with tin roofs and therefore are easily damaged by the flood water.

## Study Area Map

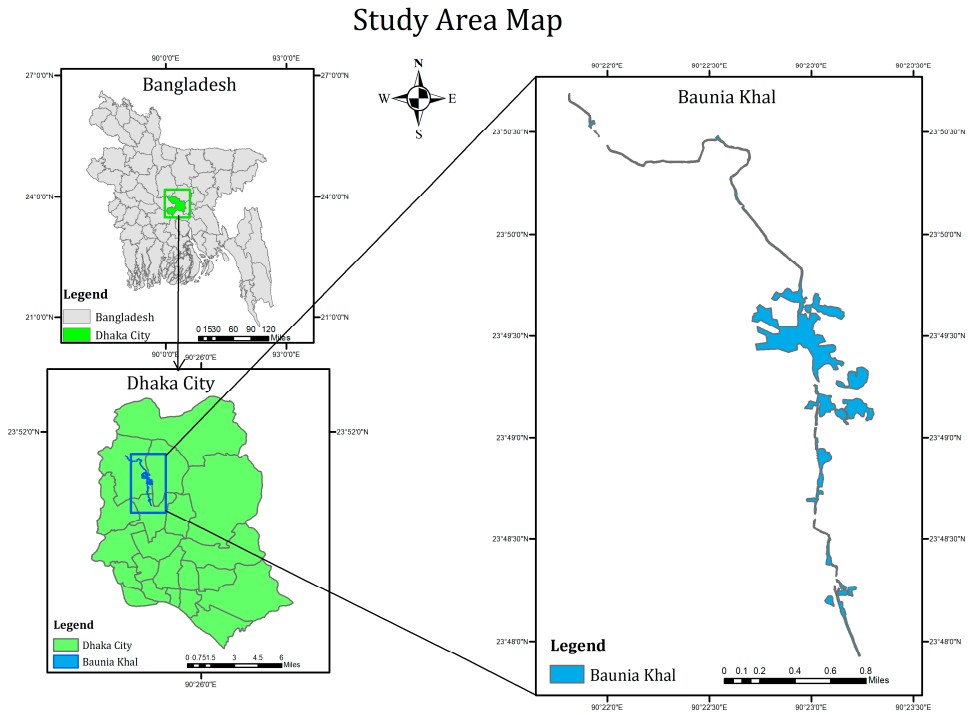

**Figure 1.** Map of the study area.

The study has been carried out to analyze the current situation of urban flooding and its extent in the Baganbari slum area. The Baunia Khal, flowing near the Baganbari slum, is a natural drainage system that overflows due to the occurrence of even short-duration rainfall. To analyze the extent of urban flooding, Intensity–Duration–Frequency (IDF) curves, spatial analysis, and peak runoffs were obtained. The 3 h rainfall data were obtained from the Bangladesh Meteorological Department for the period: 1 January 2003 to 21 October 2020. Shuttle Radar Topography Mission (SRTM) data with a resolution of 1 arc-second (30 m) were collected from the United States Geological Survey (USGS) for the year 2020. The land use and land cover data and hydrologic soil group data for the present situation in Dhaka were obtained for the same year from the European Space Agency (ESA) and the National Aeronautics and Space Administration's (NASA) Earth data. The encroachment or decrease in the watershed area of the Baunia Khal from 2004 to 2020 is measured using Google Earth images.

### 2.1. Methods for Assessing Urban Flooding Extent

The extent of urban flooding was investigated using the IDF curve, spatial analysis, and rainfall–runoff analysis. A schematic diagram of the method for investigating the urban flooding extent used in this study is shown below (see Figure 2).

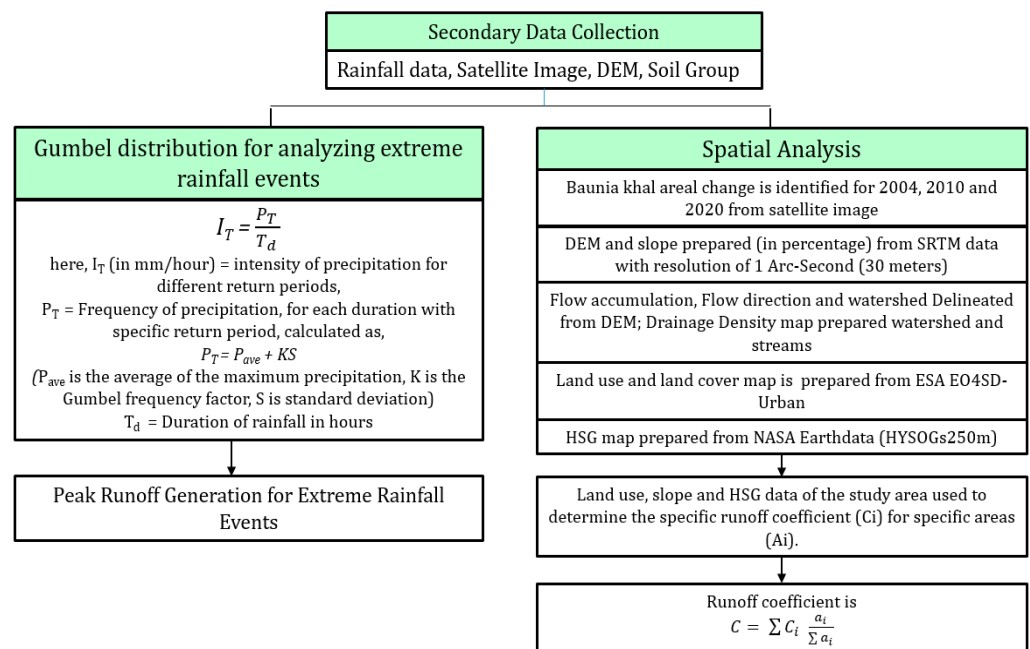

**Figure 2.** Diagram of the method for investigating urban flooding extent.

2.1.1. IDF Curve for Dhaka

The Gumbel theory of distribution is a widely used distribution theory for the analysis of Intensity–Duration–Frequency (IDF) which uses extreme events of rainfall [23,25,26]. Return intervals of 2, 5, 10, 25, 50 and 100 years for each duration period of rainfall were calculated using the Gumbel method. Frequency of precipitation, $P_T$ for each duration with a specific return period $T$ was determined by using the formula:

$$P_T = P_{ave} + KS \tag{1}$$

$P_{ave}$ is the average of the maximum precipitation of a specific duration, $T$ is the return period, and $K$ is the Gumbel frequency factor which can be specified by the following equation:

$$K = -\frac{\sqrt{6}}{\pi}\left[0.5772 + ln\left[ln\left[\frac{T}{T-1}\right]\right]\right] \tag{2}$$

Standard deviation, $S$, was determined using $P_{ave}$ and $P_i$ values, where $P_i$ denotes the individual extreme value of precipitation and $n$ is the number of recorded events. The equation is shown below:

$$S = \sqrt{\left[\frac{1}{n-1}\sum_{i=1}^{n}(P_i - P_{ave})^2\right]} \tag{3}$$

Then, the value of standard deviation, $S$, is multiplied by frequency factor $K$ which is the function of the return period and the sample size. The multiplied value of $K$ and $S$ is added with $P_{ave}$ as shown in Equation (1) in order to obtain the value of $P_T$.

The intensity of precipitation $I_T$ (in mm/hour) for different return periods was obtained from $P_T$ and $T_d$ (the duration of rainfall in hours). The equation is written below:

$$I_T = \frac{P_T}{T_d} \tag{4}$$

From the raw data, the average maximum precipitation ($P_T$) and the statistical variables (average and standard deviation) for each duration (3, 6, 12, 15, 18, 21 and 24 h) were then calculated.

### 2.1.2. Spatial Changes of Baunia Khal

Rainfall–runoff estimation for this study was carried out using the rational formula and Geographic Information System (GIS) approach. Data generation for estimating the khal area change over the past few decades was carried out using Google earth images. Using the images, the Baunia Khal map with areal increase or decrease was identified for the years 2004, 2010, and 2020. The maps were then generated in ArcGIS using image processing and spatial analysis packages.

### 2.1.3. Runoff Estimation

The peak discharge of the study area was calculated using the rational formula. The formula is written below:

$$Q = C \times I \times A \tag{5}$$

where $Q$ is peak discharge (cubic meter per hour), $C$ is the runoff coefficient, $I$ denotes rainfall intensity (meter per hour) and $A$ is the area of khal (square meter).

The land use, slope and hydrologic soil group (HSG) data of the study area were used to determine the specific runoff coefficient ($C$) for specific areas ($A$). The tables which are used to determine the runoff coefficient of the study area are given in Table A2.

The runoffs were also generated for the return period of 2, 5, 10, 25, 50 and 100 years. The runoff coefficient was also adjusted according to Dhakal, et al., 2013 [40], and rainfall intensities were collected from IDF curve values for different return periods.

For calculating the length of khal, the time of rise or time to peak was retrieved as follows,

$$T_p = \left(\frac{0.6}{t_c}\right) + \left(\frac{t_r}{2}\right) \tag{6}$$

where $T_p$ is time to peak, $t_r$ is recession time and $t_c$ is time of concentration.

According to Kirpich (1940) equation [41], time of concentration is

$$L^{0.77} = \left(\frac{t_c * S^{0.385}}{0.066}\right) \tag{7}$$

Here, $L$ is the length in meters and $S$ is the main channel slope in meter/meter [41].

The width of the khal was determined using the Simas and Hawking equation (2002) [42]. The equation is written below:

$$W^{0.594} = \frac{t_L * S^{0.15}}{0.322 * S_{nat}^{0.313}} \tag{8}$$

Here, lag time $t_L$ is in hours, watershed width is in kilometers, and slope $S$ is in m/m.

### 2.1.4. Flood Area Analysis Using Satellite Images

The flooded area of the Baganbari slum was analyzed via the synthetic aperture radar (SAR) technique using Sentinel-1B band data. A combination of images was acquired for the flooding period and non-flooding period. The image acquired during the non-flooding period or before the flooding event were called archive images, whereas the images acquired during flooding were considered crisis images [43]. According to data availability of the study area, the archive image was taken on 10 April 2020 and the crisis image was taken on 21 July 2020, after heavy rainfall occurred on 20 July 2020. Data collection was carried out from the Copernicus open access hub from the European Space Agency (ESA) in Interferometric Wide (IW) swath mode.

## 3. Results

### 3.1. Extreme Rainfall Analysis Using IDF Curve

The IDF curve, obtained using Gumbel distribution [23], showed a decrease in rainfall frequency and intensity with increased rainfall duration for each return period. Addition-

ally, higher intensity of rainfall was also found as per the increase in return periods for all rainfall durations.

Figure 3 shows the IDF curve of Dhaka for a 3 h rainfall duration interval. The highest rainfall was found within the first 3 h, and then the intensity decreased. Obviously, among the curves, the highest rainfall intensity was the 100-year return period and the lowest rainfall intensity curve was found after a 2-year return period. Interestingly, all return period curves were found to meet closely at the 24th hour.

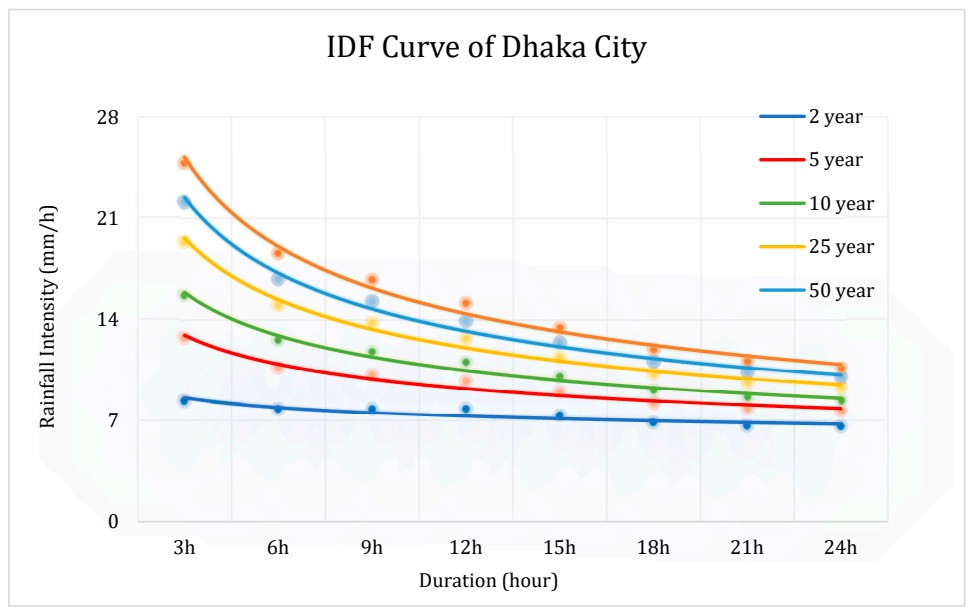

**Figure 3.** The IDF curve of Dhaka after a 3 h rainfall during intervals 2, 5, 10, 25, 50, and 100-year return periods.

### 3.2. Spatial Changes of Baunia Khal

A massive change in the area of Baunia Khal was found over the past 20 years from the GIS analysis using Google Earth data. The area of the khal was found to be 7 km$^2$ in 2004, which decreased at an alarming rate to 2.17 km$^2$ in 2010. The areal change rate was 69% from 2004 to 2010. Later, the situation worsened and the khal area is now found to be only 0.41 km$^2$. Almost 81% of the khal area was encroached and turned into urban settlement. Baunia Khal, once called "Kahor Doriya" (meaning huge waterway), is now less than 1 km$^2$, and its existence can hardly be found. Figure 4a–c show the progressive area map of the khal for 2004, 2010, and 2020.

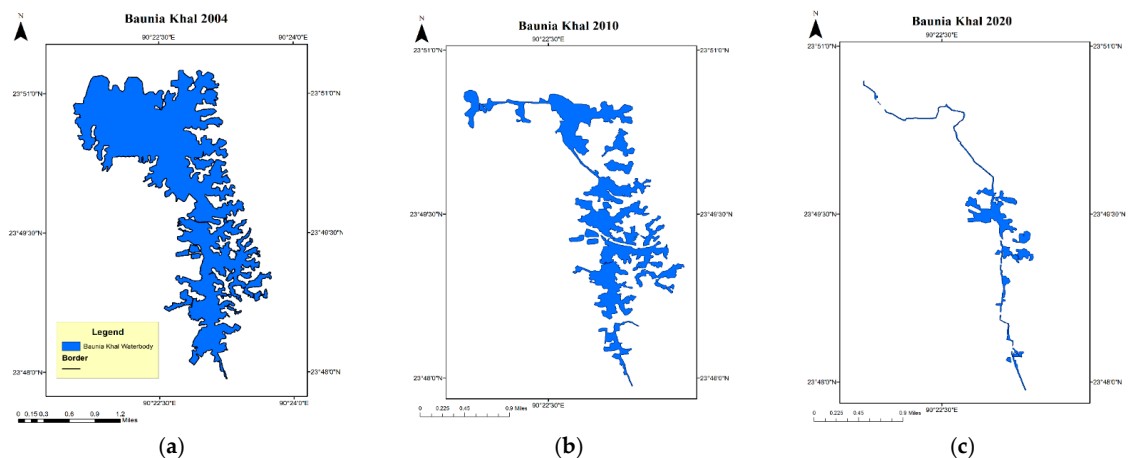

**Figure 4.** Baunia Khal conditions in 2004 (**a**), 2010 (**b**), and 2020 (**c**).

### 3.3. Flood Area Analysis from Satellite Images

The flood area was also analyzed using synthetic aperture radar (SAR) data from the sentinel 1B image. The SAR images show flooded areas as dark patches of low backscatter return which happened because of the specular reflection over a smooth surface of water. The light-colored areas in Figure 5 show rougher and more rugged terrain with signals being reflected away from flooded regions. It was found that 34.09% of the Baganbari slum was flooded during the flood that happened in the time period from 20 July 2020 to 22 July 2020. Since the area of the slum was 1.01 ha, the flooded area of the slum was 3443.1 sq m.

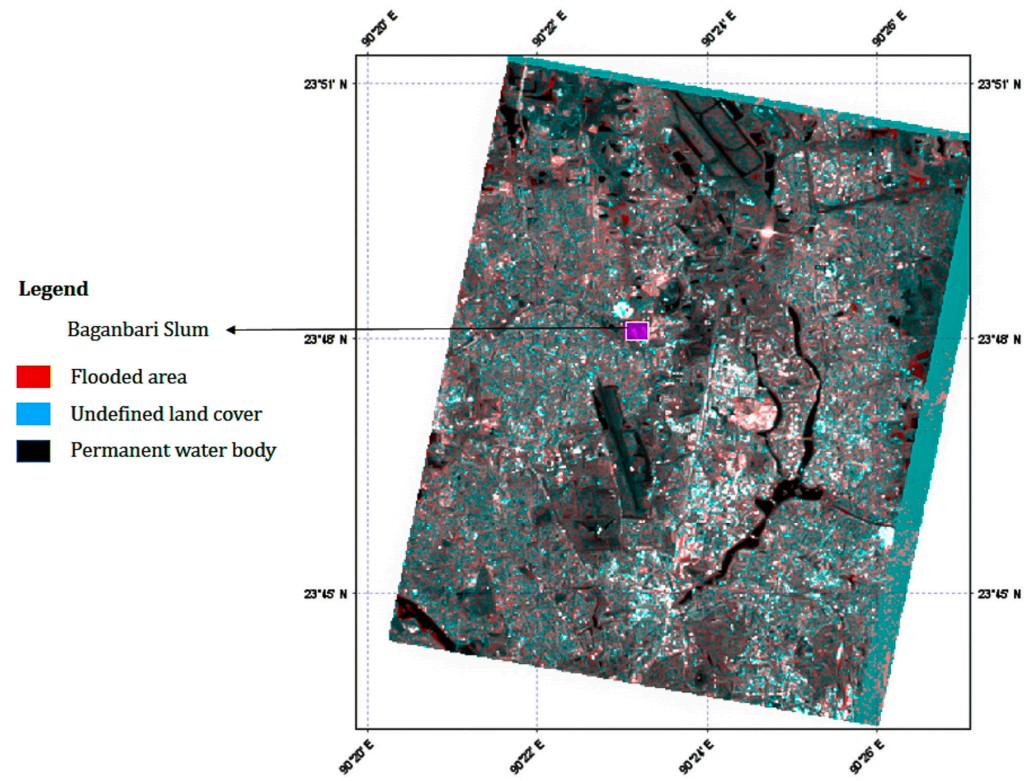

**Figure 5.** Flood area analysis of the study area.

### 3.4. Runoff Calculation

As short duration and high-intensity rainfall were frequent in the study area, the 3 h and 5-year return period of rainfall intensity, which is 12.74 mm/h was used to calculate runoff. Further, the area of the khal was found from spatial analysis of the Baunia Khal. There were seven types of land use patterns in the area: inland water; agricultural areas; other natural and semi-natural areas (savannah, grassland); artificial non-agricultural vegetated areas; industrial, commercial, public, military, private and transport units; mines, dumps and construction sites; urban fabric. The study area slope was found to be less than 2%, and soil consisted of two HSGs: group-A soil (deep sandy soil with infiltration of 7.6–11.43 mm/h), and group D (heavy plastic clays with a low infiltration of 0–1.27 mm/h).

Therefore, it turns out that:

Area A = 412,325.7 m$^2$, intensity, I = 12.74 mm/h = 0.01274 m/h, runoff coefficient was 0.76, which was generated from the land use, slope and hydrologic soil group (HSG) of the study area for the year 2020. The data from runoff coefficient, rainfall intensity and area contribute to determining the peak runoff.

Thus, Peak runoff is: Q = 66.54 m$^3$/min = 3992.30 m$^3$/h;

Therefore, the peak runoff for the flood that occurred in 2020 was found to be 3992.30 m$^3$/h. Using the Kirpich equation [41] and the Simas and Hawking equation [42], length and width of the khal were found to be 16.76 km and 20 m, respectively, whereas the present drainage length was 6.95 km, and the width was 9.5 m.

### 3.5. Peak Runoff Generation for Extreme Rainfall Events

Runoff was also computed for the rainfall intensity with return periods of 2, 5, 10, 25, 50 and 100 years, and a heightened extent of urban flooding was obtained with the increase in rainfall return periods. The runoff coefficient was adjusted for different return periods, and peak runoff was calculated accordingly.

Peak runoff was calculated for the rainfall intensity with a return period of 2, 5, 10, 25, 50 and 100 years, as shown in Figure 6. Thus, the peak runoff value was found to increase with the increase in rainfall return periods. The highest peak runoff value was found after a 3 h duration of rainfall which denotes that short-duration rainfall with high intensity caused the highest peak of runoff.

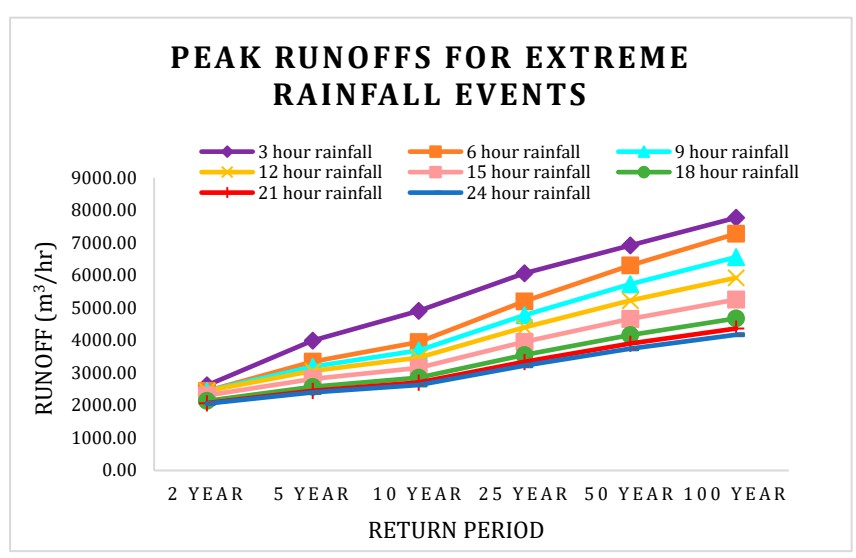

**Figure 6.** Peak runoffs for heavy-intensity rainfall with different return periods (2, 5, 10, 25, 50, and 100 years).

## 4. Discussion

Rainfall Intensity–Duration–Frequency (IDF) curves show the amount of precipitation occurring for a given time duration in Dhaka. Rainfall with a return period of 2, 5, 10, 25, 50 and 100 years generates higher intensity rainfall with extreme urban flooding events. Peak runoffs calculated from higher intensity rainfall (see Figure 6) can cause great impact in areas surrounding Baunia Khal, including the Baganbari slum. As the drainage capacity of the khal has decreased over the past few decades, heavy rainfall events such as 25, 50 and 100 years will wash away infrastructure near waterways. IDF curves are necessary to evaluate the drainage conditions of Baunia Khal. Rainfall intensity decreases simultaneously as rainfall duration increases, but intensity increases with the increase of return periods. IDFs found from the research are necessary for designing any flood control structure or predicting rainfall intensity in that region [44].

The area of the Baunia Khal has massively decreased over the past 20 years. The spatial analysis shows the decrease in the Baunia Khal area occurred from 2004 to 2020, which is recorded as 7 km$^2$ and 0.41 km$^2$, respectively. The reason behind this huge land use change is the heavy encroachment of the Baunia Khal in an illegal way through anthropogenic activities (e.g., construction and/or illegal dwelling) [39]. In the Dhaka Water Supply and Sewerage Authority (DWASA) storm water master plan 2016 [38], it is described that illegal construction of structures and heavy encroachment have happened in the Baunia Khal, creating critical challenges for storm water management. For example, the major channel for the Mirpur Zone was identified for draining storm water but land filling activity has limited the completion of the project [38].

The width of the Baunia Khal is observed to be 9.5 m in some sections of the khal, which is way less than the calculated width (20 m) required for the present situation of

the Baunia Khal. We also found that a critical condition for the Baunia Khal's existence relied on two urbanized areas (e.g., Baunia Khal: Mirpur area on its western side and Uttara at its eastern side). Thus, after construction of the road, adjoining eastern and western sides of the Baunia Khal has increased commercial activities in these two areas for which the real estate companies started to fill up the wetlands and Baunia Khal. Vigorous land filling activities in this area are the reason behind the significant areal change of the Baunia Khal [28].

The peak discharge of the study area is 3992.30 m$^3$/hour found from this research study. Due to the encroachment of the study area, peak discharge from a short duration of rainfall causes urban flooding and inundates the Baganbari slum [45]. Sentinel-1B image analysis shows 34.09% of the area of the Baganbari slum was flooded on 21 July 2020 after a heavy rainfall event occurred on 20 July 2020. Though Baganbari slum dwellers mentioned that 100% of the area of the slum was flooded during that period, the low resolution of the Sentinel images could detect only 34% of the flooded slum area (3443.1 sq m.). A short duration of high-intensity rainfall with improper management and capacity of the khal is the main reason behind the high runoff discharge to the khal [46]. Further, intense rainfall, watershed infiltration, evapotranspiration, and groundwater discharge also increase the peak flow of the water body [47].

Flooding is a major hazard in both rural and urban areas, although the economic and social impacts caused by flooding are higher in urban areas than in rural [9]. Urban flooding situations in Baunia Khal are debilitating and disruptive, having a profound impact on the livelihood of Baganbari slum dwellers. Urban flooding causes financial loss and affects the sanitation condition and health of the dwellers. It brings about disruption in income-generating activities, by increasing travel costs and time, decreasing customers at stalls and making it difficult for slum dwellers to go to work. Hence, increasing the area of the Baunia Khal and frequent excavation is necessary to prevent urban flooding. Additionally, public awareness among the slum dwellers is required to not dispose of domestic waste in the khal and obstruct its flow.

## 5. Conclusions

Living alongside a khal in Dhaka without any embankment brings about enormous suffering due to flooding during the monsoon period. Both short- and long-term impacts are visible in the lives of people who live beside the khal, such as Baganbari slum dwellers. The Baganbari slum is situated next to Baunia Khal, which is regularly flooded even after a short duration of rainfall. It is vigorously encroached, and illegal settlements are developed on the watershed area of Baunia Khal, which has led to the narrowed-down situation of this khal. Additionally, the khal is severely polluted by domestic waste from the slum dwellers, which blocks the flow of the khal and increases the risk of urban flooding. As a result, the condition of Baunia Khal is very poor and alarming. Once called "Kahor Doriya", the Baunia Khal is in critical condition as its drainage area and capacity have decreased tremendously. Therefore, the research findings from this study emphasize the destructive urban flooding events of the Baunia Khal watershed for which comprehensive flood management strategies are required to ensure a sustainable and resilient future for the urban areas in Dhaka.

**Author Contributions:** Z.S. investigated the proposed research and conducted urban flooding evaluation and analysis; and is the primary on the manuscript. S.K.B. and J.H.R. proposed the study and contributed to conceptualizing the project (S.K.B.), interpreting the processes in general (J.H.R.) as Z.S.'s advisor. All authors have read and agreed to the published version of the manuscript.

**Funding:** Publication of this article was partially funded by the University of Idaho—Open Access Publishing Fund. This work is also supported by J.H.R's project funded by the USDA National Institute of Food and Agriculture, Hatch project 1023305. Any opinions, findings, or recommendations expressed in this publication are those of the authors and do not necessarily reflect the view of USDA.

**Conflicts of Interest:** The authors declare no conflict of interest.

## Appendix A

**Table A1.** Pair-wise Ranking of Water-Related Problems in Baganbari Slum.

|  | Urban Flooding | Waste Management | Water Quality | Water Price | Sanitation | Ranking |
|---|---|---|---|---|---|---|
| Urban flooding | Urban flooding | Urban flooding | Urban flooding | Urban flooding | Urban flooding | 1st |
| Waste management | Urban flooding | Waste management | Water quality | Waste management | Waste management | 3rd |
| Water quality | Urban flooding | Water quality | Water quality | Water quality | Water quality | 2nd |
| Water price | Urban flooding | Waste management | Water quality | Water price | Sanitation | 5th |
| Sanitation | Urban flooding | Waste management | Water quality | Sanitation | Sanitation | 4th |

## Appendix B

**Table A2.** Runoff Coefficient (C).

| LULC | Soil Group A | | | Soil Group B | | | Soil Group C | | | Soil Group D | | |
|---|---|---|---|---|---|---|---|---|---|---|---|---|
| Slope: | <2% | 2–6% | >6% | <2% | 2–6% | >6% | <2% | 2–6% | >6% | <2% | 2–6% | >6% |
| Forest | 0.08 | 0.11 | 0.14 | 0.1 | 0.14 | 0.18 | 0.12 | 0.16 | 0.20 | 0.15 | 0.20 | 0.25 |
| Meadow | 0.14 | 0.22 | 0.3 | 0.2 | 0.28 | 0.37 | 0.26 | 0.35 | 0.44 | 0.30 | 0.40 | 0.50 |
| Pasture | 0.15 | 0.25 | 0.37 | 0.23 | 0.34 | 0.45 | 0.30 | 0.42 | 0.52 | 0.37 | 0.50 | 0.62 |
| Farmland | 0.14 | 0.18 | 0.21 | 0.16 | 0.21 | 0.28 | 0.20 | 0.25 | 0.34 | 0.24 | 0.29 | 0.41 |
| Res. 1 acre | 0.22 | 0.26 | 0.29 | 0.24 | 0.28 | 0.34 | 0.28 | 0.32 | 0.40 | 0.31 | 0.35 | 0.46 |
| Res. 1/2 acre | 0.25 | 0.29 | 0.32 | 0.28 | 0.32 | 0.36 | 0.31 | 0.35 | 0.42 | 0.34 | 0.38 | 0.46 |
| Res. 1/3 acre | 0.28 | 0.32 | 0.35 | 0.3 | 0.35 | 0.39 | 0.33 | 0.38 | 0.45 | 0.36 | 0.4 | 0.5 |
| Res. 1/4 acre | 0.3 | 0.34 | 0.37 | 0.33 | 0.37 | 0.42 | 0.36 | 0.40 | 0.47 | 0.38 | 0.42 | 0.52 |
| Res. 1/8 acre | 0.33 | 0.37 | 0.4 | 0.35 | 0.39 | 0.44 | 0.38 | 0.42 | 0.49 | 0.41 | 0.45 | 0.54 |
| Industrial | 0.85 | 0.85 | 0.86 | 0.85 | 0.86 | 0.86 | 0.86 | 0.86 | 0.87 | 0.86 | 0.86 | 0.88 |
| Commercial | 0.88 | 0.88 | 0.89 | 0.89 | 0.89 | 0.89 | 0.89 | 0.89 | 0.90 | 0.89 | 0.89 | 0.90 |
| Street | 0.76 | 0.77 | 0.79 | 0.8 | 0.82 | 0.84 | 0.84 | 0.85 | 0.89 | 0.89 | 0.91 | 0.95 |
| Parking | 0.95 | 0.96 | 0.97 | 0.95 | 0.96 | 0.97 | 0.95 | 0.96 | 0.97 | 0.95 | 0.96 | 0.97 |
| Distributed area | 0.65 | 0.67 | 0.69 | 0.66 | 0.68 | 0.7 | 0.68 | 0.72 | 0.75 | 0.69 | 0.72 | 0.75 |
| Cropland | 0.14 | 0.18 | 0.22 | 0.16 | 0.21 | 0.28 | 0.20 | 0.25 | 0.34 | 0.24 | 0.29 | 0.41 |
| Forest | 0.08 | 0.11 | 0.14 | 0.10 | 0.14 | 0.18 | 0.12 | 0.16 | 0.20 | 0.15 | 0.20 | 0.25 |
| Grassland | 0.15 | 0.25 | 0.37 | 0.23 | 0.34 | 0.45 | 0.30 | 0.42 | 0.52 | 0.37 | 0.50 | 0.62 |
| Mixed vegetation | 0.14 | 0.22 | 0.30 | 0.20 | 0.28 | 0.37 | 0.26 | 0.35 | 0.44 | 0.30 | 0.40 | 0.50 |
| Artificial surfaces | 0.33 | 0.37 | 0.40 | 0.35 | 0.39 | 0.44 | 0.38 | 0.42 | 0.49 | 0.41 | 0.45 | 0.54 |

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
