# Peer review of "Assessing Urban Flooding Extent of the Baunia Khal Watershed in Dhaka, Bangladesh"

_water, doi:10.3390/w15061183_

Round 1

Reviewer 1 Report

To further improve the text, I suggest the following changes in the manuscript.
Abstract: Abstract should be written in concise. I would suggest listing only some of the most important results to justify the implications and conclusions of the study.
The background of an introduction should be revised accordingly.
The introduction is very good. It doesn't reflect the goal; please rewrite it again, it is suggested to include some latest reference. 
Objectives of this study must be included at end of introduction part. 
I highly recommended to authors, if possible, please modify the figure with good quality images. 
The economic intuition behind the results are missing. The author/s should revise the discussion part. The result should be supported with recent studies.
What is contribution of this work to existing literature? 
It has been observed that the authors have used old references and ignored the latest studies. So it is suggested to add recent references. Please check reference section some references are missing.The policy implications also required elaboration. The implications should go along with the results and the course of action should be discussed in this part. In some places, some grammatical errors are found that need to be fixed.

please read following articles and cite properly, 

DOI: https://doi.org/10.15244/pjoes/134292

10.5004/dwt.2020.25119

Author Response

See the attached pdf file. 

Reviewer 2 Report

The manuscript assesses the urban flooding issues in the Baunia khal area of Dhaka, Bangladesh. The work utilizes the IDF curves to calculate extreme rainfall-runoff for a small watershed in Dhaka, Bangladesh. The topic is relevant to this journal. 

There are many scopes to improve the manuscript. The paper will be presented to international readers upon publication who do not have prior information about the study area. In many cases, it seems like the authors assumed the reader would know local facts. For example, the local name of the study area is not translated; connectivity between Mirpur, Uttara, and the study area is unknown to the reader, among others. Some more specific comments are listed below.

Specific comments:

Line 111-115: What are the timeline of the used SRTM topo data and the land use/land cover data?

Line 133: Please check the equation.

Line 185: Please add the rainfall depth and duration of the crisis event.

Line 199: Figure 3: What are the two points near 15hr duration?

Line 203: Previously, line 113 indicated using European Space Agency data for land use. However, line 203 indicated the use of Google Earth for land use. Please, for consistency, write about the data source properly.

Line 208: What is the meaning of the term “Kahor Doriya”?

Line 220: Figure 5: The full satellite image tile seems irrelevant to the manuscript. The study area is too small to see in this image.

Line 227: How is the land use pattern of the study area?

Line 230: This line is not clear. The authors are using 3 hrs duration-5yrs return period rainfall analysis and calculate the runoff to be 3,992.30 m3/hour. It is surprising to see the peak runoff for 2020 be the same. Also, please provide a reference for the 2020 flood peak. 

Line 254: The authors did not provide enough information on the infrastructures near waterways. What are the design periods of these infrastructures? What are those infrastructures in this discussion?

Line 270: What is the actual width of the khal? If 9.75m is underestimated, does it mean the actual width is much higher?

Line 273-277: These lines are hard for readers to connect. The mentioned locations are not shown on the maps. A reader cannot connect the relationship between these localities and the khal under discussion if s/he is not local.

Line 282-285: These lines are not clearly explained: "Though Baganbari slum dwellers mentioned ... resolution of Sentinel image indicates". What is the total flooded area (in sq. m)?

Author Response

See the attached pdf file

Round 2

Reviewer 1 Report

accept

Author Response

Thank you for accepting our manuscript. Again, we appreciate your time. 

Reviewer 2 Report

I would like to thank the authors for their response to my comments. The responses are adequate to most comments. However, some responses are not quite enough to justify some information presented in this paper. I am listing them here.

1. Line 238: It seems like a 3hr-5 year return period flood peak is 3992 m3/hr. As the authors are calculating this value to be the same flood peak as the 2020 event, it appears that the authors are concluding the 2020 event as a 3hr - 5 yr return period event.

The response to the comments from the authors indicates that the actual rainfall depth for the 2020 event is unknown.  What is the basis for considering it as a 3hr and 5-year return period event?

2. If the authors do not want to add the land use pattern to the paper, it is advised to add a couple of lines describing the land use pattern of the study area only (not the whole of Dhaka city). It is important to add information on how the authors have come to a runoff coefficient of 0.76 for the study area.

3. The conclusion needs to be more focused. Some sentences like the following are not covering the main focus of the paper.

"Additionally, the khal is severely polluted by waste and sewerage water disposed all along from Mirpur. Baganbari slum dwellers also dispose their domestic waste in the khal since they don’t have any waste disposal system. As a result, the condition of Baunia khal is very poor and alarming."

These sentences are about pollution and illegal activities, but neither of them is the main focus of this paper.

Another sentence in conclusion is: "Therefore, the research finding will provide useful insights for water mangers and/or city planners to plan out their water management exercises by minimizing potential flood hazards in the study area.

- It is expected that the authors reemphasize the study findings in the conclusion section.

Author Response

Reviewer #2

I would like to thank the authors for their response to my comments. The responses are adequate to most comments. However, some responses are not quite enough to justify some information presented in this paper. I am listing them here.

>>Thanks for your constructive review. We respond to your comments as below.

  1. Line 238: It seems like a 3hr-5 year return period flood peak is 3992 m3/hr. As the authors are calculating this value to be the same flood peak as the 2020 event, it appears that the authors are concluding the 2020 event as a 3hr - 5 yr return period event.

The response to the comments from the authors indicates that the actual rainfall depth for the 2020 event is unknown.  What is the basis for considering it as a 3hr and 5-year return period event?

>> We describe the basis for considering 3hr and 5-year return period rainfall event (See Line 232 to 234.

The 3-hour duration and 5-year return period of rainfall is considered for this study due to the short duration and high-intensity rainfall being frequent in the study area.

  1. If the authors do not want to add the land use pattern to the paper, it is advised to add a couple of lines describing the land use pattern of the study area only (not the whole of Dhaka city). It is important to add information on how the authors have come to a runoff coefficient of 0.76 for the study area.

>> Information about the land use, HSG, and slope of the study area is added in the revised manuscript (See Line 234 – 240).

  1. The conclusion needs to be more focused. Some sentences like the following are not covering the main focus of the paper.

"Additionally, the khal is severely polluted by waste and sewerage water disposed all along from Mirpur. Baganbari slum dwellers also dispose their domestic waste in the khal since they don’t have any waste disposal system. As a result, the condition of Baunia khal is very poor and alarming."

These sentences are about pollution and illegal activities, but neither of them is the main focus of this paper.

Another sentence in conclusion is: "Therefore, the research finding will provide useful insights for water mangers and/or city planners to plan out their water management exercises by minimizing potential flood hazards in the study area."

- It is expected that the authors reemphasize the study findings in the conclusion section.

>> Conclusion session is revised as per the reviewer’s suggestion. Thank you.